# Growth-Inhibitory Effect of Chitosan-Coated Liposomes Encapsulating Curcumin on MCF-7 Breast Cancer Cells

**DOI:** 10.3390/md18040217

**Published:** 2020-04-17

**Authors:** Mahmoud Hasan, Kamil Elkhoury, Nabila Belhaj, Cyril Kahn, Ali Tamayol, Muriel Barberi-Heyob, Elmira Arab-Tehrany, Michel Linder

**Affiliations:** 1LIBio, Université de Lorraine, F-54000 Nancy, France; mahmoud.hasan@univ-lorraine.fr (M.H.); kamil.elkhoury@univ-lorraine.fr (K.E.); nabila_Belhaj@cargill.com (N.B.); cyril.kahn@univ-lorraine.fr (C.K.); 2Department of Biomedical Engineering, University of Connecticut, Storrs, CT 06269, USA; 3CRAN, Université de Lorraine, CNRS, F-54000 Nancy, France; muriel.barberi@univ-lorraine.fr

**Keywords:** breast cancer, MCF-7, liposomes, encapsulation, chitosan, curcumin

## Abstract

Current anticancer drugs exhibit limited efficacy and initiate severe side effects. As such, identifying bioactive anticancer agents that can surpass these limitations is a necessity. One such agent, curcumin, is a polyphenolic compound derived from turmeric, and has been widely investigated for its potential anti-inflammatory and anticancer effects over the last 40 years. However, the poor bioavailability of curcumin, caused by its low absorption, limits its clinical use. In order to solve this issue, in this study, curcumin was encapsulated in chitosan-coated nanoliposomes derived from three natural lecithin sources. Liposomal formulations were all in the nanometric scale (around 120 nm) and negatively charged (around −40 mV). Among the three lecithins, salmon lecithin presented the highest growth-inhibitory effect on MCF-7 cells (two times lower growth than the control group for 12 µM of curcumin and four times lower for 20 µM of curcumin). The soya and rapeseed lecithins showed a similar growth-inhibitory effect on the tumor cells. Moreover, coating nanoliposomes with chitosan enabled a higher loading efficiency of curcumin (88% for coated liposomes compared to 65% for the non-coated liposomes) and a stronger growth-inhibitory effect on MCF-7 breast cancer cells.

## 1. Introduction

Curcumin is a yellow natural polyphenolic compound extracted from turmeric root (*Curcuma longa*). Curcumin has been widely used in traditional medicine due to its pharmacological effects such as its antioxidant, anticancer, anti-inflammatory, and anti-microbial activities [1,2]. The anticancer properties of curcumin are owed to its ability to target multiple cellular and molecular cancer pathways, such as the nuclear factor kappa-light-chain-enhancer of activated B cells (NF-kB), tumor protein p53, phosphatase and tensin homolog (PTEN), mitogen-activated protein kinase (MAPK), and the microRNAs (miRNAs) network [3]. Recent studies have indicated that the anticancer properties of curcumin could be exerted via targeting various miRNAs, such as the upregulation of miR-7, miR-9, miR-21 and miR-181a, and the downregulation of miR34a and miR-200c [3,4,5].

However, the poor oral bioavailability of curcumin limits its therapeutic efficacity [6,7]. In fact, its poor solubility in an aqueous medium and weak stability in alkaline pH conditions or gastrointestinal fluids restrict its availability to cross into the blood circulation by oral administration. To overcome such limitations, curcumin has been previously encapsulated in polymeric nanoparticles [8], liposomes [9], biodegradable microspheres [10] and hydrogels [11,12].

Liposomes are some of the oldest delivery carriers that are widely used to deliver bioactive agents to cells and tissues while protecting them from physiological barriers [13,14]. When encapsulated in liposomes, bioactive agents can be protected from stomach digestion and absorbed in substantial amounts in the gastrointestinal tract [15]. The main constituents of liposomes are the amphiphilic phospholipids, which are also the main constituents of cell membranes [16,17]. The amphiphilic property of phospholipids allows liposomes to self-seal and deliver hydrophobic drugs in aqueous media. Nanoparticles such as nanoliposomes, which are in the range of 100 to 200 nm, possess an extended bloodstream circulation time due to their size, which is large enough that it helps them in avoiding selective uptake in the liver, but small enough that it helps them in avoiding mechanical filtration by the spleen. Additionally, nanoliposomes possess improved intracellular accumulation and localization in the tumor area, and they can passively target tumor cells through the enhanced permeability and retention (EPR) effect [18,19].

Lecithin nanoliposomes can be extracted and produced from salmon, soya and rapeseed [20,21,22]. These types of nanoliposomes contain a high percentage of omega-3 long-chain polyunsaturated fatty acids (n-3 LC-PUFAs), such as docosahexaenoic acid (DHA, 22:6 n-3) and eicosapentaenoic acid (EPA, 20:5 n-3) [23]. These n-3 LC-PUFAs have anticancer effects which are poorly understood, but it has been reported that they are based on the generation of reactive oxygen species [24], alteration in gene expression [25], increase in drug transport across the cell membrane [24,26], induction of apoptosis [27], lipid peroxidation [24], and the modulation of cellular proliferation [28] and differentiation [29].

Chitosan has been well considered for use in bio-adhesive drug delivery systems, in order to improve the bioavailability of drugs by increasing their time in residence at the absorption site. As highlighted by Kotze et al. and Rengel et al., chitosan is a hydrophilic polymer with good biocompatible and biodegradable properties, leading to a low toxicity level [30,31]. It has been used previously as a coating material for many types of nanoparticles to reduce their cytotoxicity and improve their mucoadhesive properties [32,33,34,35]. Due to the presence of amino groups, chitosan is polycationic. To achieve a prolonged and controlled release, a positively charged chitosan coating is generally formed on the surface of negatively charged liposomes via an ionic interaction [35,36,37].

Here, we characterized the physicochemical properties of different formulations of chitosan-coated lecithin nanoliposomes extracted from salmon, rapeseed and soya. In addition, we also investigated the effect of curcumin-loaded nanoliposomes on the viability of Michigan Cancer Foundation-7 (MCF-7) breast cancer cells.

## 2. Results and Discussion

### 2.1. Nanoliposomes Size and ζ-potential

The particle size, polydispersity index (PDI) and ζ-potential of different formulations were measured immediately after preparation and presented in Figure 1. The average diameter of a chitosan-coated nanoliposome was 317, 358 and 318 nm for nanoparticles from soya, salmon and rapeseed lecithins, respectively. We observed that salmon nanoliposomes coated with chitosan had the largest diameter. This can be explained by the interaction between the lipid and chitosan, which caused a greater bilayer expansion in the nanoliposomes rich in unsaturated fatty acids compared to the ones rich in saturated fatty acids [38].

The PDI of soya, rapeseed and salmon nanoliposomes coated with chitosan was 0.23, 0.23 and 0.22, respectively. This indicates that particle size was well controlled, with a narrow dispersity, since the PDI value is <0.3. According to the mean values and PDIs, there was no significant difference between the breadths of distribution of the samples. We found that the PDI of the curcumin-loaded nanoparticles is slightly greater than the unloaded curcumin nanoparticles.

Uncoated nanoliposomes showed negative ζ-potential values, which became positive after coating with chitosan. The ζ-potential increased from -43.3, -40.9 and -43.5 mV for soya, salmon and rapeseed liposomes, respectively, to 60.9, 66.2 and 66.8 mV after coating with chitosan, respectively. The ζ-potential of uncoated nanoliposomes was negative, probably due to the anionic fractions of lecithin [20]. The increase in the surface charge of chitosan-coated nanoliposomes was attributed to the increase in the positively-charged amino groups of chitosan molecules, proving that the nanoliposome coating was successfully achieved [39]. Regarding the stability of the formulations, no significant variation in particle size or charge was observed during storage periods of 30 days at 4 °C and 37 °C, which suggests that the liposomes can be stored without lyophilization for a minimum of 1 month without showing any changes in their properties.

### 2.2. Encapsulation Efficiency of Curcumin

The encapsulation efficiency of curcumin was 87.15, 88.61 and 88.72% in rapeseed, soya and salmon chitosan-coated nanoliposomes, respectively. Compared to our previous study [23], these results indicate that the encapsulation efficiency of curcumin significantly increases when liposomes are coated with chitosan.

### 2.3. Membrane Fluidity

The fluidity of the liposomes reflects the order and dynamics of the phospholipids’ alkyl chains in the vesicle’s bilayer. The fatty acid (FA) composition tunes the membrane’s fluidity level. Membrane fluidity decreases when saturated FAs are present, due to an increase in the packing between the phospholipids, whilst unsaturated FAs increase membrane fluidity by reducing the packing between the phospholipids [40].

To recognize the action of curcumin and chitosan on membrane fluidity, it is necessary to understand the behavior of curcumin and chitosan with respect to solution composition variation. According to our previous study [23], membrane fluidity depends on the lipid composition of nanoliposomes, as a lower membrane fluidity was found for rapeseed and salmon liposomes compared to soya liposomes. This can be explained by the higher proportion of PUFAs with short chains found in soya liposomes compared to rapeseed and salmon liposomes.

As shown in our previous results [23], curcumin decreased the membrane fluidity of all nanoliposomes, as its presence can weaken the hydrophobic interactions among acyl chains. 

In this study, as shown in Table 1, the presence of chitosan also reduced the membrane fluidity of nanoliposomes, as this new layer around the liposome was probably incorporated within the membrane bilayer, thus increasing the rigidity of the bilayers and decreasing the movement of the FA chains. Consequently, the membrane bilayer fluidity decreased and the motional freedom of the phosphate group was reduced [41].

### 2.4. Morphology of the Liposomes

Small unilamellar vesicles (SUV) can be observed in the TEM images of nanoliposomes that were prepared via sonication followed by high-pressure homogenization (Figure 2a). A small quantity (10%) of oil droplets in each formulation, in the form of nanoemulsions, can be observed. Figure 2b shows a chitosan contrasting band surrounding the nanoliposomes. The TEM images confirm the size results measured via the dynamic light scattering (DLS) technique.

### 2.5. Growth-Inhibition by Real-time Cell Analysis

The growth-inhibition of uncoated and chitosan-coated nanoliposomes, that were empty or loaded with curcumin, was measured via an impedance-based analysis method. Their composition and concentration effects on MCF-7 cells’ growth was investigated. Curcumin concentrations of 12 µM or higher showed a significant effect on cell index (CI), whereas the 5 µM curcumin concentration showed no significant difference (Figure 3). Reports on the anti-tumor activity of curcumin were based on two opposite mechanisms: inhibiting anti-apoptosis proteins and activating the pro-apoptosis proteins [42].

Some studies suggest that curcumin acts as an antioxidant and prevents reactive oxygen species (ROS) production [43,44]. Other studies suggest that curcumin induces ROS production at high concentrations and quenches ROS production at low concentrations [45,46]. The antioxidant mechanism mediates NF-κB-suppressive effects and the pro-oxidant mechanism mediates apoptotic effects [45]. Other studies report that tumor cells show the preferential uptake of curcumin compared to normal cells and that the toxicity clearly increases with increasing curcumin uptake; therefore, this advantage makes it an attractive agent for cancer therapy. Among various factors that are responsible for higher curcumin uptake in tumor cells against normal cells could be their difference in membrane structure, protein composition and bigger size [47,48,49].

Curcumin, in its free form, has limited clinical efficacy due to being weakly absorbed in the gastrointestinal tract. The nanoliposome encapsulation of curcumin could improve its systemic administration. For certain concentrations, nanoliposomes can have anticancer effects. According to our previous study [23], nanoliposomes produced from vegetable lecithin can significantly decrease the cell proliferation of cancer cells at a concentration of 20 µM. Interestingly, salmon-derived nanoliposomes can have a growth-inhibitory effect with a lower concentration (5 µM). This could be due to the difference in their lecithin composition, as salmon lecithin possess a higher polyunsaturated fatty acid composition, mainly eicosapentaenoic acid (EPA) and Docosahexaenoic acid (DHA), which exert anticancer effects [50,51,52]. This study evaluated the in vitro anti-tumor activity of curcumin-loaded nanoliposomes on the MCF-7 cancer cell line. The results show that free curcumin of 5 µM has a lower growth-inhibitory impact on MCF-7 cells compared to curcumin loaded in salmon nanoliposomes (Figure 3). This suggests that a synergic MCF-7 growth-inhibitory effect exists between salmon nanoliposomes and curcumin.

The coating of liposomes extends their in vivo life span and permits the accumulation of liposomes in the target sites. Because of its bioadhesive and permeation-enhancing properties, chitosan has received substantial attention as a drug delivery system [30]. We observed that chitosan-coated liposomes inhibited more tumor growth compared to uncoated liposomes, especially at high concentrations (Figure 3).

The results presented in Figure 3 indicate that growth-inhibition increased for all types of lecithin coated with chitosan. These data indicate the increased sensitivity of MCF-7 cells toward curcumin-loaded chitosan-coated liposomes. Rapeseed and soya liposomes induced the proliferation of MCF7 cells at low concentrations, but they increased the growth-inhibition at higher concentrations. Salmon liposomes were toxic to MCF-7 cells at different concentrations. Moreover, the coating of liposomes with chitosan led to the growth of MCF7 cells at low concentrations, whereas it significantly increased growth-inhibition at higher concentrations. 

## 3. Materials and Methods

Salmon lecithin, from *Salmo salar*, was obtained by enzymatic hydrolysis. The lipids were extracted by the use of an enzymatic process without any organic solvent, as described previously [53], whilst rapeseed and soya lecithins were commercial lecithins from the Solae Europe SA society. Chitosan (deacetylation degree up to 75%), curcumin, boron trifluoride (14% in methanol), acetonitrile (≥99.9%), chloroform (≥99.9%), methanol (≥99.9%) and hexane (≥99.9%) were all purchased from Sigma-Aldrich (Saint-Quentin-Fallavier, France) and Fisher Scientific (Illkirch, France). Acetic acid (≥99.8%) was supplied by Prolabo-VWR. All the organic solvents were analytical grade reagents.

### 3.1. Preparation of Chitosan-coated Liposomes

To prepare the liposome solution, 1.5 g of lecithin and 10 mg curcumin were dissolved in ethanol, then a thin lipid film was formed on the wall of the flask using a Rotavapor by completely evaporating the ethanol under vacuum, then the lipid film was hydrated with 47.5 mL of distilled water and the suspension was agitated using a magnetic stirrer for 4 h in an inert nitrogen atmosphere. In total, 0.5 g of chitosan and 0.5 mL of acetic acid were then added to the solution. The suspension was mixed again for 4 h under the same conditions. After that, the solution was first sonicated at 40 kHz for 5 min (1s on, 1s stop) in an ice bath and then homogenized using a high-pressure homogenizer (EmulsiFlex-C3, Sodexim SA, Muizon, France) in aliquots of 50 mL under a pressure of 1500 bar for 7–8 cycles.

### 3.2. Physicochemical Characterization and Stability Analysis

The mean diameter, particle size distribution and ζ-potential of vesicles were determined upon the dilution of the samples (1:200) using the DLS technique, by employing a Zetasizer Nano ZS (Malvern Instruments Ltd., Worcestershire, UK).

Empty and curcumin-loaded chitosan-coated liposomes were stored in a drying room at 4 °C and 37 °C for five weeks. The mean particle size, PDI and ζ-potential of all formulations were analyzed every 3 days.

### 3.3. Encapsulation Efficiency of Curcumin

The curcumin’s concentration was determined via reverse-phase HPLC (Shimadzu, Kyoto, Japan), using a Zorbex SB-C18 column (5µm, 4.6mm×250mm). The mobile phase consisted of 2% acetic acid, methanol and acetonitrile, at a ratio of 30:5:65. The elution was carried out with a flow rate of 0.5 mL/min. A twenty microliter aliquot of the solution was injected into the HPLC at ambient temperature and a wavelength of 425 nm was used for detection. The retention time for curcumin was about 8.49 min and its linearity was obtained in the range of 2 to 20 µg/mL. The total drug content of the suspensions was determined by dissolving chitosan-coated liposomes in methanol and measuring the drug content with HPLC. First, the entrapment efficiencies of the curcumin were determined by the centrifugation of the nanoliposome at 1000 g for 10 min, to separate the free curcumin, and then the supernatant was re-centrifuged at 200,000 g for 4 h to separate the unloaded curcumin. The free drug in the supernatant was detected by HPLC. The total drug content of the suspensions was also determined using a similar procedure. The encapsulation efficiency was calculated as:EE % = ((total drug – free drug)/total drug) × 100

### 3.4. Membrane Fluidity

The membrane fluidity of all samples was measured by fluorescence anisotropy measurements. TMA–DPH was used as the fluorescent probe. The measurement was carried out according to the method described by Maherani et al. [54]. In brief, the solution of TMA–DPH (1 mM in ethanol) was added to the liposome suspension to reach a final concentration of 4 µM and 0.2 mg/mL for the probe and the lipid, respectively. The mixture was lightly stirred for 1 h at ambient temperature in the dark. Then, 180 µL of the solution was distributed into each well of a 96-well black microplate. The fluorescent probe was vertically and horizontally oriented in the lipid bilayer. The fluorescent intensity of the samples was measured with a Tecan INFINITE® 200 PRO (Grödig, Austria) equipped with fluorescent polarizers. Samples were excited at 360 nm and emission was recorded at 430 nm under constant stirring at 25 °C. The Magellan 7 software was used for data analysis. The polarization value (P) of TMA–DPH was calculated using the following equation:$$P=\frac{I_{II}-GI_{┴}}{I_{II}+2GI_{┴}}$$
where *I_II_* is the fluorescent intensity parallel to the excitation plane, I_⊥_ the fluorescent intensity perpendicular to the excitation plane, and G is the factor that accounts for transmission efficiency. Membrane fluidity was defined as 1/*P*. The results were measured in triplicate.

### 3.5. Transmission Electron Microscopy (TEM) 

The morphology of the nanoliposomes was monitored via TEM, using a negative staining method, as previously described [21]. In brief, the liposomal formulation was diluted in distilled water (1:10 ratio) and then mixed with 2% ammonium molybdate solution with a ratio of 1:1. The mixture was reserved at room temperature for 3 min. Then, one drop was placed and dried on a Formvar carbon-coated copper grid (200 mesh, 3 mm diameter HF 36). Then, the morphology of the nanoliposomes was examined using a Philips CM20 TEM equipped with an Olympus TEM CCD camera at 200 kV.

### 3.6. In Vitro Evaluation of the Anti-cancer Activity of Encapsulated Curcumin

#### 3.6.1. Cell Culture

Human breast cancer MCF-7 cells (NCCS Pune) were cultured in an RPMI 1640 medium without phenol red (Gibco™, ThermoFisher Scientific, Grand Island, NY, USA), supplemented with 10% (v/v) fetal bovine serum, 1% penicillin/streptomycin and 2mM l-Glutamine, at 37°C in a humidified atmosphere of 5% CO_2_.

#### 3.6.2. Evaluation of Drug Toxicity

The in vitro cellular effect on MCF-7 cells of the curcumin-loaded nanoliposome was studied using the xCELLigence system (Roche Diagnostics GmbH, Mannheim, Germany), as described previously [23]. In brief, 1 x 10^4^ cells per well were cultured overnight in 96-well E-Plates^TM^. Three different concentrations of curcumin, encapsulated in nanoliposomes (5, 12 and 20 µM), were mixed with the culture medium. The concentrations of nanoliposomes and chitosan corresponding to 5, 12, and 20 µM of curcumin are presented in Table 2. The xCELLigence system (real-time cell analyzer single plate, RTCASP^®^) allows the real-time monitoring of cell proliferation based on impedance measurement. The technology uses specific 96-well cell culture E-Plates^TM^ with bottoms covered with microelectrodes as an electrical impedance cell sensor. The analysis is based on the measurement of electrical impedance created by attached cells across the high-density electrode array coating the bottom of the wells [55]. The impedance value is automatically converted to a dimensionless parameter called a cell index, which is defined as the relative change in electrical impedance created by the attached cells. As a quantitative measure of cellular status, the CI value represents the extent of the cell-covered area, and it is directly related to cell number, cell proliferation, cell size and morphology, cell viability and attachment forces [56,57].

Control wells were composed of different concentrations of curcumin solubilized in ethanol, and curcumin-free nanoliposomes. Impedance was measured every 15 min to continuously monitor cell growth.

## 4. Conclusions

In summary, we extracted lecithin from natural marine (salmon) and vegetable (rapeseed and soya) sources. These formulations were used to produce liposomes, which were loaded with curcumin and coated with chitosan. Liposomal formulations were characterized to determine their morphological and physicochemical properties. Coating nanoliposomes with chitosan increased their size, changed their charge from negative to positive and achieved a higher encapsulation efficiency of curcumin. Growth-inhibition assays on MCF-7 breast cancer cells showed an increase in growth-inhibition when concentrated liposome formulations were loaded with curcumin or coated with chitosan. As mentioned before, curcumin, in its free form, has limited clinical efficacy as it is weakly absorbed in the gastrointestinal tract. The liposomes increase the bioavailability of curcumin and the chitosan coating increases its circulation time. These findings show the potential of marine- and plant-derived coated nanoliposomal formulations as controlled drug delivery systems for breast cancer treatment. The next steps should be to check if this growth-inhibitory effect is caused by the inhibition of the cells’ proliferation or the cells’ death, and to study these formulations on other breast cancer cell types, such as SKBR3 and MDA-MB231, before progressing to in vivo studies.

## Figures and Tables

**Figure 1 marinedrugs-18-00217-f001:**
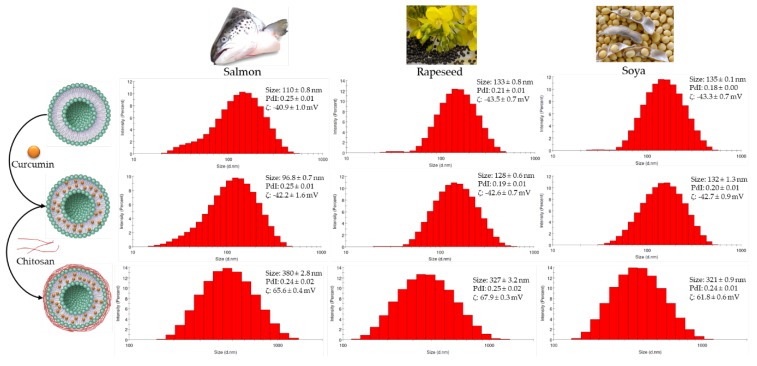
Schematic and physicochemical characterization of salmon, rapeseed and soya nanoliposomes, curcumin-loaded nanoliposomes and curcumin-loaded nanoliposomes coated with chitosan.

**Figure 2 marinedrugs-18-00217-f002:**
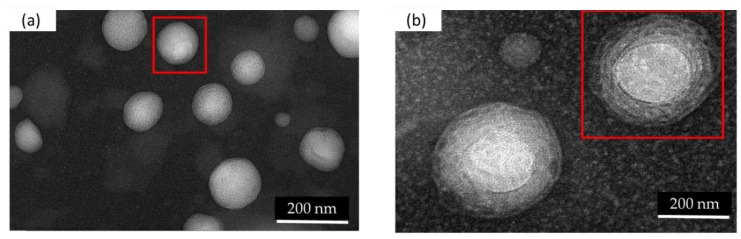
Transmission Electron Microscopic (TEM) images of a curcumin-loaded nanoliposome before (**a**) and after (**b**) coating with chitosan.

**Figure 3 marinedrugs-18-00217-f003:**
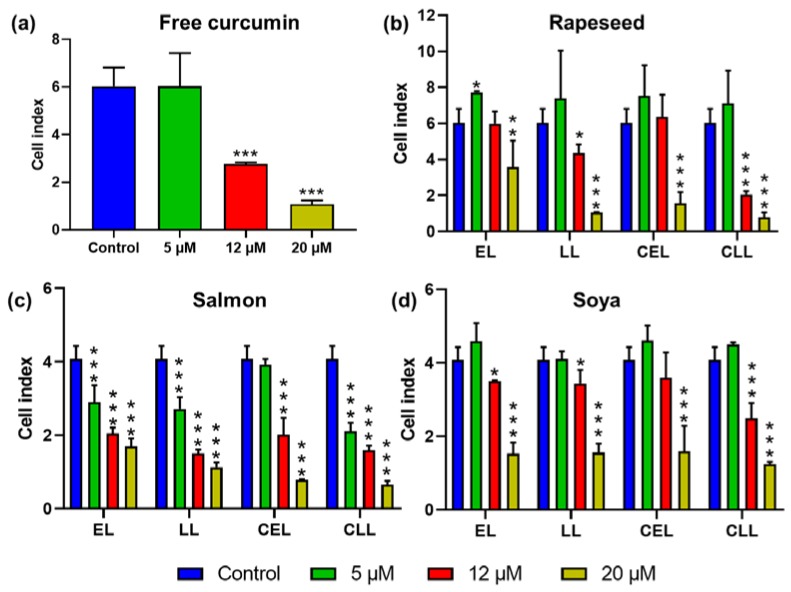
MCF-7 cell index kinetics exposed to (**a**) free curcumin, (**b**) rapeseed, (**c**) salmon and (**d**) soya, empty liposomes (EL), curcumin-loaded liposomes (LL), chitosan-coated empty liposomes (CEL) and chitosan-coated curcumin-loaded liposomes (CLL). Cell index values were recorded after 96 h of liposome exposure. The reported data are the means of six replicates. Parametric data were analyzed using a one-way (a) and two-way (b–d) ANOVA followed by Dunnett’s test. Significance was indicated as * (*p* < 0.05), ** (*p* < 0.01) and *** (*p* < 0.001).

**Table 1 marinedrugs-18-00217-t001:** Membrane fluidity of the chitosan-coated liposomes (each value represents the mean of triplicates).

Sample	Membrane Fluidity
Soya CEL	3.21 ± 0.10
Soya CLL	2.56 ± 0.10
Salmon CEL	2.70 ± 0.10
Salmon CLL	2.62 ± 0.20
Rapeseed CEL	3.21 ± 0.10
Rapeseed CLL	2.71 ± 0.10

CEL: chitosan-coated empty liposomes; CLL: chitosan-coated curcumin-loaded liposomes.

**Table 2 marinedrugs-18-00217-t002:** Concentrations of nanoliposomes and chitosan corresponding to 5, 12, and 20 µM of curcumin.

Concentration	Curcumin	Nanoliposomes	Chitosan
1	20 µM	1.1 mg/mL	0.37 mg/mL
2	12 µM	0.66 mg/mL	0.22 mg/mL
3	5 µM	0.28 mg/mL	0.09 mg/mL

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
