# Peer review of "Growth-Inhibitory Effect of Chitosan-Coated Liposomes Encapsulating Curcumin on MCF-7 Breast Cancer Cells"

_marinedrugs, 2020, doi:10.3390/md18040217_

Round 1

Reviewer 1 Report

The manuscript describes physiochemical characteristics and effects on cancer cells of a liposome formulated to encapsulate curcumin, and stabilized by chitosan coatings.  

The following clarifications are recommended:

English language issues and grammatical inconsistencies should be addressed. 

Abstract: 

The abstract would benefit by making more quantitative comparisons.  Instead of saying "this group has the highest effect", be clear about how much more effect compared to the control groups.  Same with loading efficiency and cellular effects.  

In reading the article, I am not sure the tests proved anything about cytotoxicity.  Rather, growth of the cells was monitored by impedance methods.   Any effect could only be called "Growth-inhibitory" unless proof of cell death was obtained by other methods.   Would recommend modifying any claims, including the title, that include "Cytotoxic". 

Intro: 

line 39, for the unfamiliar reader, what are the listed miRNAs implicated in?  When you say curcumin "targets", does it block those pathways? enhance them?  

Results and discussion: 

For fatty acid analyses and lipid classes, there are no figures and tables that would justify "it can be seen by ..... ".  How much info does this add to the main study.  Could it be included as a supplement rather? The primary focus of the study is liposomes, not sources of lipids.   

Same for solubility.  No table of values or way to easily visualize and compare results.  Again, the focus of the article is on liposomes, not curcumin solubility.  Consider removing, condensing into other discussion, or providing as supplement. 

Figure 1 does not appear to be called out in the text.  

Cell index needs to be clarified.  How is this calculated? does higher index mean more cells? Is this particular measure specific to the instrument used? 

It is unclear what the relevance of figure 3 is to this study on liposome delivery?  

line 203, obtained instead of provided

line 205 commercial (without s)

methods: 

section 3.4, how were the lipoomes and curcumin mixed? what type of stirrer?  

Clarify the stability studies. Empty and loaded and coated liposomes were stored at two different temps?  Body temperature for 5 weeks?  Are results and discussion inluded for the stability over time? 

Conclusions repeats the results of the studies, but would be more informative to the reader if this paragraph puts these results into context of clinical significance.  How would this be delivered as a clinical therapy?  implanted? ingested?  What are the next steps that should occur to get this to the next step.  

Author Response

  1. Reviewer: English language issues and grammatical inconsistencies should be addressed.

Response:We thank the reviewer for the comment.We have revised the manuscript again to address English language issues and grammatical inconsistencies.

  1. Reviewer: The abstract would benefit by making more quantitative comparisons. Instead of saying "this group has the highest effect", be clear about how much more effect compared to the control groups. Same with loading efficiency and cellular effects.

Response:We thank the reviewer for the comment.  we added and highlighted quantitave comparasions to the abstract.

  1. Reviewer: ‎In reading the article, I am not sure the tests proved anything about cytotoxicity. Rather, growth of the ‎cells was monitored by impedance methods.   Any effect could only be called "Growth-inhibitory" unless proof ‎of cell death was obtained by other methods.  Would recommend modifying any claims, including the title, ‎that include "Cytotoxic".
    Response:We thank the reviewer for the comment. We changed cytotoxic effect to growth-inhibitory effect throughout the manuscript.
  2. Reviewer: ‎line 39, for the unfamiliar reader, what are the listed miRNAs implicated in? When you say curcumin "targets", does it block those pathways? enhance them?
    Response:We thank the reviewer for the comment. We added which miRNAs curcumin upregulates and whiche ones it downregulates.
  3. Reviewer:For fatty acid analyses and lipid classes, there are no figures and tables that would justify "it can be seen by ..... ". How much info does this add to the main study.  Could it be included as a supplement rather? The primary focus of the study is liposomes, not sources of lipids.  Same for solubility.  No table of values or way to easily visualize and compare results.  Again, the focus of the article is on liposomes, not curcumin solubility.  Consider removing, condensing into other discussion, or providing as supplement.
    Response:We thank the reviewer for the comment. We took the advice of the reviewer to focus on liposomes and not lipids into consideration. So, we removed fatty acids, lipid classes, and solubility sections and added a section about liposomes’ membrane fluidity.
  4. Reviewer:Figure 1 does not appear to be called out in the text.
    Response:We thank the reviewer for the comment. We modified this and called Figure 1 out in line 82.
  5. Reviewer:Cell index needs to be clarified.How is this calculated? does higher index mean more cells? Is this particular measure specific to the instrument used?
    Response:We thank the reviewer for the comment. The clarification of cell index was added to section 3.6.2.
  6. Reviewer: ‎It is unclear what the relevance of figure 3 is to this study on liposome delivery?
    Response:We thank the reviewer for the comment. We removed old figure 3 and transformed old figure 4 into figure 3 by removing the ethanol only related study and adding the free curcumin related histograms.
  7. Reviewer: ‎line 203, obtained instead of provided. line 205 commercial (without s).
    Response:We thank the reviewer for the comment. We have fixed those errors.
  8. Reviewer: ‎section 3.4, how were the lipoomes and curcumin mixed? what type of stirrer?
    Response:We thank the reviewer for the comment. Section 3.4 (now 3.1) has been modified to be include more informations about liposomes and curcumin mixing and the type of stirrer.
  9. Reviewer: ‎Clarify the stability studies. Empty and loaded and coated liposomes were stored at two different temps? Body temperature for 5 weeks? Are results and discussion inluded for the stability over time?
    Response:We thank the reviewer for the comment. We have studied the stability of liposomes at 4°C and 37°C to check if their physichochemical properties would change over time, and we did not notice any change in their size or charge during a storage period of 30 days. A text was added in lines 99-102 to better clarify this point.
  10. Reviewer: ‎Conclusions repeats the results of the studies, but would be more informative to the reader if this paragraph puts these results into context of clinical significance. How would this be delivered as a clinical therapy? implanted? ingested?  What are the next steps that should occur to get this to the next step.
    Response:We thank the reviewer for the comment. We have modified the conclusion to include the significance of this study and the nex steps required.

Reviewer 2 Report

1. The role of ethanol in figure 3 should be explained in more detail, and also why the authors chose to use ethanol to perform these experiments.

2. Some comment regarding the liposomes long-term stability (size, potential, non-aggregation...) should be made, including storage possibilities. 

Author Response

  1. Reviewer: ‎The role of ethanol in figure 3 should be explained in more detail, and also why the authors ‎chose to use ethanol to perform these experiments‎.
    Response:We thank the reviewer for the comment. Since we used ethanol to dilute curcumin as it has a low solubility in water, we wanted to prove that the cytotoxicity towards cells was caused by curcumin and not by ethanol, that is why we studied ethanol as acontrol. However, based on reviewrs #1 advice, we removed Figure 3.
  2. Reviewer: ‎Some comment regarding the liposomes long-term stability (size, potential, non-aggregation...) should be made, including storage possibilities.
    Response:We thank the reviewer for the comment. We added these comments in line 99-102.

Reviewer 3 Report

My comments in document.

Author Response

  1. Reviewer: ‎Put each reference with each effects respectively.
    Response:We thank the reviewer for the comment. We have added the references with each effects respectively‎ in lines 65-67.
  2. Reviewer: ‎It is neccesary more discussion in the parts of fatty acids, lipid classes and solubility.
    Response:We thank the reviewer for the comment. We took the advice of the reviewer #1 to focus on liposomes and not lipids into consideration. So,‎we removed fatty acids, lipid classes, and solubility sections and added a section about liposomes’ ‎membrane fluidity.‎
  3. Reviewer: ‎Give one reference in line 139.
    Response:We thank the reviewer for the comment. Since we removed the solubility section we removed this sentence as well.
  4. Reviewer: ‎What do you can said about size fron thos TEM images?
    Response:We thank the reviewer for the comment. We added a discussion part about the size in the TEM section (lines 148-149).
  5. Reviewer: ‎Explain more about cell index (CI) and explain the relationship between impedance and CI..
    Response:We thank the reviewer for the comment. The detailed explanation about cell index and impedance was added to section 3.6.2.

Round 2

Reviewer 1 Report

Comments and recommendations have been addressed sufficiently.